# Use of Rv0222-Rv2657c-Rv1509 Fusion Protein to Improve the Accuracy of an Antibody ELISA for Extra-Pulmonary Tuberculosis in Humans

**DOI:** 10.3390/pathogens10070828

**Published:** 2021-06-30

**Authors:** Yingyu Chen, Pan Ge, Kailun Zhang, Jie Xiang, Li Zhang, Ian D. Robertson, Aizhen Guo

**Affiliations:** 1State Key Laboratory of Agricultural Microbiology, Huazhong Agricultural University, Wuhan 430070, China; chenyingyu@mail.hzau.edu.cn (Y.C.); gepan345543@gmail.com (P.G.); kelen91@aliyun.com (K.Z.); 2College of Veterinary Medicine, Huazhong Agricultural University, Wuhan 430070, China; 3Hubei International Scientific and Technological Cooperation Base of Veterinary Epidemiology, National Animal Tuberculosis Para-Reference Laboratory (Wuhan), Huazhong Agricultural University, Wuhan 430070, China; I.Robertson@murdoch.edu.au; 4Wuhan Jinyintan Hospital, Wuhan 430023, China; xiangj2005@126.com (J.X.); zhangli080806@163.com (L.Z.); 5School of Veterinary Medicine, Murdoch University, Murdoch, WA 6150, Australia

**Keywords:** tuberculosis, extra-pulmonary tuberculosis, pulmonary tuberculosis, diagnosis, fusion protein

## Abstract

(1) Background: Tuberculosis (TB) in humans is a serious chronic epidemic disease caused by *Mycobacterium tuberculosis* (*M. tb*). The diagnosis of TB, especially extra-pulmonary TB (EPTB), is difficult. Isolation of *M. tb* from culture has a low sensitivity in patients with TB and an even lower sensitivity in cases of EPTB. Although Xpert MTB/RIF assays and serological tests are more sensitive than the above tests, they still lack sensitivity for EPTB diagnosis. (2) Methods: To improve the accuracy of TB diagnosis, a Rv0222-Rv2657c-Rv1509 fusion protein based iELISA was constructed, the diagnosis of TB, pulmonary TB (PTB) and EPTB was then evaluated. Sera of 40 TB patients including 14 with PTB, 14 with EPTB and 12 with no information about the form of TB, and five pneumonia patients were investigated. (3) Results: The sensitivity of the ELISA in TB, PTB and EPTB patients was 80% (95% CI: 64.4, 90.9%), 85.7% (95% CI: 57.2, 98.2%) and 92.8% (95% CI: 66.1, 99.8%), respectively, with a specificity of 70% (95% CI: 53.5, 83.4%). Both the sensitivity and specificity with this fusion protein were higher than for CFP10/ESAT6 (used as reference antigen) fusion protein (71.4%; 95% CI: 41.9, 91.6%, and 67.5%; 95% CI: 50.9, 81.4%), respectively, in cases of EPTB. All pneumonia patients’ sera tested negative in both ELISAs. (4) Conclusion: use of these new fusion proteins as antigens in serological assays has the potential to improve the diagnosis of all forms of TB in humans, especially EPTB.

## 1. Introduction

Tuberculosis (TB) in humans remains a major health problem in developing countries and is responsible for the highest mortality globally of all infectious diseases [1]. In 2018 the WHO estimated that 10 million people fell ill with TB resulting in more than 1.2 million deaths worldwide with the most cases being located in India [1]. Extra-pulmonary TB (EPTB) constitutes between 15% and 20% of all cases of TB in humans with this number reportedly increasing over the last decade [2]. The diagnosis of TB, especially EPTB, remains challenging as there is no single method suitable for its diagnosis. Culturing the causative agent, *Mycobacterium tuberculosis* (*M. tb*), from a specimen obtained from the patient and detection of acid-fast Bacillus (AFB) in smears of sputum or tissue are two conventional methods used to diagnose tuberculosis; however, in all cases of TB they have a low reported sensitivity (20–60%) [3]; which is even lower for the diagnosis of EPTB [2,4]. In order to improve the sensitivity of TB diagnosis, the Xpert MTB/RIF assay was developed in 2006, however it has the disadvantage of high cost and low sensitivity of 52.5% for EPTB with low bacterial loads and is not capable of distinguishing viable from non-viable *M. tb* in patients [5]. Serological tests are widely used in the diagnosis of tuberculosis due to their convenience, ease of implementation and speed [6,7]. Over the past decade a number of antigens, such as Rv2026c, Rv 2421c [8], 45-kDa [9], Ag85 [10], CFP-10, ESAT6, and Rv3615c [11], have been targeted by serodiagnostic tests; however, serological confirmation of EPTB is rare. Others have demonstrated that incorporating a range of antigens, such as cocktail and fusion proteins, can improve the sensitivity of antibody ELISA tests compared with ELISAs only using a single antigen [12,13]. 

In the current study, we tested specific antigens from the M. tb genome using the web tools (http://genome.tbdb.org/ access date: 15 June 2018) and the *M. tb* H37Rv database (tuberculist.epfl.ch access date: 15 June 2018) to improve the sensitivity and specificity of an antibody ELISA for TB and EPTB. Here we found Rv0222, Rv1509 and Rv2657 C were with highly immunogenic and had potential use for TB diagnosis.

## 2. Results

### 2.1. Generation of Fusion Antigens

The similarity of cloned sequences and those in the *M. tb* genome (GenBank accession no: AL123456.3) was > 99% at the nucleic acid level and 100% identity at the amino acid level. SDS PAGE results showed that Rv0222-Rv2657c-Rv1509 fusion protein was successfully expressed and purified (Figure 1).

### 2.2. Rv0222-Rv2657c-Rv1509-iELISA for TB Diagnosis

Using a cut-off value of 0.30 the AUC on the Rv0222-Rv2657c-Rv1509-iELISA for the diagnosis of TB (all types) was 0.8 ± 0.1 (Area ± Std. Error) (95% CI: 0.7, 0.9, *p* < 0.0001) (Figure 2A). A total of 32 of the 40 serum samples from TB patients were classified as positive compared with sera from 12 of the 40 healthy controls resulting in a calculated sensitivity of 80% (95% CI: 64.4, 90.9), and a specificity of 70% (95% CI: 53.5, 83.4) (Figure 2B).

Using a cut-off value of 0.5 the AUC of the CE-iELISA was 0.8 ± 0.04 (95% CI: 0.8, 0.9, *p* < 0.0001) (Figure 2C). A total of 31 of the sera from TB cases were classified as positive compared with 11 of the healthy controls. The sensitivity was calculated at 77.5% (95% CI: 61.5, 89.2); and the specificity of 72.5% (95% CI: 56.1, 85.4) (Figure 2D). 

All five pneumonia sera tested negative using both Rv0222-Rv2657c-Rv1509-iELISA and CE-iELISA, which indicated that both Rv0222-Rv2657c-Rv1509-iELISA and CE-iELISA can distinguish TB and pneumonia patients.

### 2.3. Use of the Rv0222-Rv2657c-Rv1509-iELISA to Diagnose PTB

Using a cut-off value of 0.3 the AUC of Rv0222-Rv2657c-Rv1509-iELISA was 0.8 ± 0.07 (95% CI: 0.6, 0.9, *p* < 0.001) for the diagnosis of the 14 cases of PTB (Figure 3A). Twelve of the samples from the 14 patients with PTB were classified as positive compared with 12 of 40 samples from healthy controls (sensitivity of 85.7% (95% CI: 57.2, 98.2) and specificity of 70% (95% CI: 53.5, 83.4) (Figure 3B).

Using a cut-off value of 0.6 the AUC of the CE-iELISA was 0.8 ± 0.04 (95% CI: 0.7, 1.0, *p* < 0.001) (Figure 3C). Ten of 14 patients with PTB patients were classified as positive along with 8 of 40 healthy controls. The sensitivity of the test was estimated as 71.4% (95% CI: 41.9, 91.6) and the specificity 80% (95% CI: 64.4, 90.9) (Figure 2D).

### 2.4. Use of the Rv0222-Rv2657c-Rv1509-iELISA to Diagnose EPTB

Using a cut-off value of 0.3 the AUC of Rv0222-Rv2657c-Rv1509-iELISA was 0.8 ± 0.1 (95% CI: 0.7, 1.0, *p* < 0.01) for the diagnosis of EPTB (Figure 4A). Thirteen of 14 patients with EPTB were classified as positive compared to 12 of 40 healthy controls. The sensitivity of the i-ELISA in this component of the study was estimated to be 92.8% (95% CI: 66.1, 99.8), and the specificity 70% (95% CI: 53.5, 83.4) (Figure 4B).

Using a cut-off value of 0.5 the AUC of the CE-iELISA for diagnosing EPTB was 0.8 ± 0.1 (95% CI: 0.7, 1.0, *p* < 0.001) (Figure 4C). Ten of 14 samples from patients with EPTB had a positive antibody level compared with 13 of 40 healthy controls. The sensitivity and specificity of the test to diagnose EPTB was estimated to be 71.4% (95% CI: 41.9, 91.6) and 67.5% (95% CI: 50.9, 81.4), respectively (Figure 4D).

The results of the study are summarized in Table 1.

## 3. Discussion

The diagnosis of TB, in particular EPTB, is challenging [14]. Culture of *M. tb* in samples of sputum, broncho-alveolar lavage, and/or tissue biopsy is considered the gold standard for the diagnosis of active TB. However as *M. tb* is slow-growing, culture is time-consuming, requiring at least one month to obtain results, and has a low sensitivity [15]. Xpert and Xpert Ultra *M. tb*/RIF are new techniques recommended by the WHO to confirm a diagnosis of TB (since 2013) [5]. Although they are regarded as advancements for the diagnosis of tuberculosis because of their faster and simpler operation than conventional methods, there is lower sensitivity in EPTB and the tests are expensive, limiting their use and availability in developing countries [16]. In culture-positive patients, the sensitivity of the Xpert Ultra for the diagnosis of TB has been estimated at 83.7% (95% CI, 68.7, 92.7) and for Xpert as 67.4% (95% CI, 51.3, 80.5). In contrast the sensitivity has been estimated as 52.5% (95% CI, 45.4, 59.6) and 34.0% (95% CI, 27.6, 41.1) for Xpert Ultra and Xpert, respectively for the diagnosis of EPTB, compared with only 21.5% (95% CI, 16.2, 28.0) for bacterial culture [17]. T-SPOT, another broadly applied interferon-gamma release assay used in clinics, has been found to have a relatively higher sensitivity (78.4%) and specificity (59.0 to 93.0%) for TB compared with traditional culture, although a low accuracy in EPTB has been reported [18]. This highlights the need for a more sensitive test for the diagnosis of TB, especially in patients with EPTB.

Serological tests based on detecting circulating antibody, especially IgG, have been widely studied and used in the diagnosis of TB [19]. *M. tb* contains more than 4000 antigens [20], and of these antigens some, such as CFP10, ESAT6, Rv2026c, Rv2421c, Rv3403c, have been widely used in the serological diagnosis of TB [8,21,22,23]. The presence of Rv0222, Rv2657c and Rv1509 have previously been reported to be strongly associated with PTB [6,21], although their use in the diagnosis of EPTB has rarely been reported to our knowledge.

As use of a single dominant protein-based antigen has resulted in a poor diagnostic capacity for TB, serological assays using fusion proteins as antigens has the potential to increase the accuracy for the diagnosis of TB [24]. In this study, we combined Rv0222, Rv2657c and Rv1509 as a fusion protein and tested this in serum collected from patients with TB, PTB and EPTB along with a similar number of healthy controls. The study found that when this fusion protein was used as the antigen in the i-ELISA a sensitivity of 80% (95% CI: 64.4, 90.9%), 85.7% (95% CI: 57.2, 98.2%) and 92.8% (95% CI: 66.1, 99.8%) was found in serum from patients with TB, PTB and EPTB, respectively (Table 2). Although fusion antigen CFP10/ESAT6 has previously been found to be an excellent immunodiagnostic antigen for the diagnosis of TB [22,23], in the current study we found this antigen had a lower sensitivity in sera from all three groups of patients with TB than Rv0222-Rv2657c-Rv1509; although it did have a slightly higher specificity in sera from healthy controls in TB and PTB testing. For EPTB using fusion antigen CFP10/ESAT6 had a lower specificity than when Rv0222-Rv2657c-Rv1509 fusion antigen was used (Table 2). This indicated that Rv0222-Rv2657c-Rv1509 fusion protein has potential value in diagnostic assays for not only detecting cases of TB and PTB, but also in cases with EPTB.

Although our study showed promising results with the use of Rv0222-Rv2657c-Rv1509 fusion protein as an antigen in an i-ELISA for the diagnosis of TB, PTB and EPTB, there were some limitations with the study. A limited sample size, lack of consideration of latent TB infection, and the inability to distinguish between patients with lung cancer and/or pneumonia from patients with TB requires investigation in the future. It is recommended that further studies are conducted using Rv0222-Rv2657c-Rv1509 as the antigen in an i-ELISA to evaluate and overcome these limitations.

## 4. Materials and Methods

### 4.1. Ethics Statement

Serum samples were collected from 40 healthy volunteers and 40 TB patients. The cases of TB had been confirmed by experienced physicians from the Wuhan Jinyintan Hospital. The sampling had been approved by the Ethical Review of Biomedical Research Involving Human Subjects issued by the National Health and Family Planning Commission of The People’s Republic of China and approved by the Ethics Committee of Wuhan Medical Treatment Center (#2015001). The study was explained to all participants and written informed consent obtained prior to the study.

### 4.2. Serum Samples and Patients

All sampled individuals were confirmed HIV negative through testing at the Wuhan Medical Treatment Center. Of the 40 patients with TB, 14 had pulmonary tuberculosis (PTB), 14 had EPTB and there was no information about the form of TB in the remaining 12. The PTB cases were identified through the presence of clinical symptoms including the presence of fever, cough and expectoration, and confirmed with imaging examination (chest X-ray), bronchoscopy, and the presence of acid-fast bacilli on sputum smears and the culture of *M. tb* from the sputum. The diagnosis of EPTB was confirmed by clinical and radiographical findings, B-ultrasound, CT, Magnetic Resonance Imaging, positive antibody test by TB IgG/IgM Rapid Test Kit (Beijing Genesee Biotech, Inc., Beijing, China) in serum, pleural fluid and ascites and cerebrospinal fluid, positive acid-fast bacilli or the culture of *M. tb* from lymph nodes, pleural effusion, ascetic fluid or cerebrospinal fluid, and biopsy of the disease organs. TB patients’ information were listed in Table 2. As pneumonia and tuberculosis are easily confused in clinical diagnosis, five non-TB pneumonia patients’ sera were used for analytic specific test.

The samples from the healthy controls were obtained from student volunteers from Huazhong Agricultural University and all had no prior history or diagnosis of TB. These controls were test negative on the tuberculin skin test (induration area < 5 mm), displayed no characteristic symptoms of tuberculosis, and had a negative IFN-γ release assay by T-SPOT.TB (Oxford Immunotec, Ltd., Oxford, UK). 

### 4.3. Cloning, Expression and Purification of Target Proteins

Recombinant CFP10/ESAT6-pET28a was stocked in our laboratory. Genomic DNA of *M. tb* H37Rv strain was kindly provided by the Chinese Center for Disease Control and Prevention (CCDC, Beijing, China). 

Rv0222, Rv2657c and Rv1509 genes were amplified from the genome of *M. tb* H37Rv by PCR and fused with a linker through Gene Splicing by Overlapping Extension PCR (SOE-PCR) as previously described [25]. The primers used are listed in Table 3. The accuracy of the insertion of Rv0222-Rv2657c-Rv1509 fusion gene into the recombinant pET-28a was determined by digesting restrictively and sequenced by Sangon Biotech, Shanghai, China. Fusing genes were cloned into the expression vector pET-28a. The recombinant plasmid was sequenced and expressed in *E. coli* BL21 (DE3). Fusion protein was purified by Ni-NTA agarose chromatography (Qiagen, Chatsworth, CA, USA) after extraction.

### 4.4. Indirect ELISA (iELISA)

Purified proteins Rv0222-Rv2657c-Rv1509 and CFP10/ESAT6 (CE, used as the reference antigen, which was regarded as effective antigen for TB diagnosis in many researches [26,27]) were diluted to a final concentration of 1000 ng/mL in coating buffer (0.05 M Na_2_CO_3_-NaHCO_3_, pH 9.6), and 0.1 mL of the diluted Rv0222-Rv2657c-Rv1509 protein was added to each well of a 96-well plate. 0.1 mL of diluted CFP10/ESAT6 protein was added to each well of another plate and used as a control. Plates were then covered with adhesive plastic and incubated overnight at 4 °C. After washing three times with phosphate buffered saline with 0.05% Tween-20 (PBST), the plates were blocked with 5% skimmed milk for 1 h at 37 °C. After washing three times with PBST, 1:100 diluted serum was added to each well for 35 min at 37 °C. After washing, 100 μL horseradish peroxidase (HRP)-conjugated goat anti-human IgG (H + L) was added to each well and the plates incubated for 35 min at 37 °C. Each plate was then washed five times before 100 μL TMB (3,3′,5,5′-tetramethylbenzidine) substrate solution was added to each well for 10 min at room temperature. The stop solution (2NH_2_SO_4_) (100 μL) was then added to each well to stop the reaction. The absorbance was then read at OD 650 nm. 

### 4.5. Statistical Analyses 

The data were analyzed using a two-tailed unpaired T-Test and an ANOVA. *p* values < 0.05 were considered to be significant. Cut-off values and corresponding test sensitivity and specificity were calculated through ROC curve analysis and assessing the area under the curve (AUC) using GraphPad Prism 6.0 (GraphPad software, San Diego, CA, USA) assuming that 40 cases were positive for TB and the 40 healthy controls were negative for TB. Confidence Intervals were calculated as previously reported [28].

## Figures and Tables

**Figure 1 pathogens-10-00828-f001:**
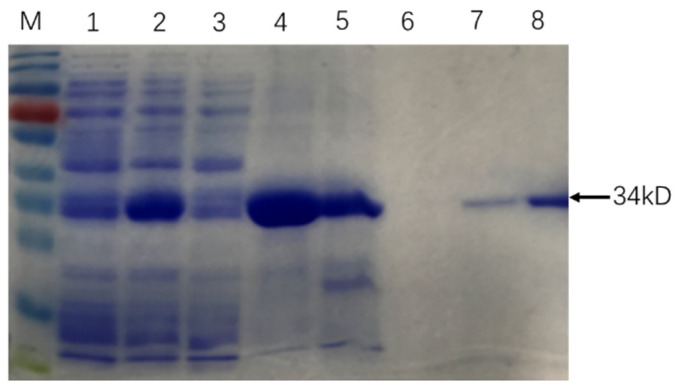
Purification of recombinant Rv0222-Rv2657c-Rv1509 protein. The arrow showed the protein Rv0222-Rv2657c-Rv1509. M: Protein Ladder; Lane 1: protein expression before induction; Lane 2: protein expression after induction; Lane 3: protein expressed in supernatant; Lane 4: protein expressed in precipitation; Lane 5: Protein in effluent; Lane 6: protein in binding buffer; Lane 7: protein in elution buffer; Lane 8: purified Rv0222-Rv2657c-Rv1509 protein after ultrafiltration.

**Figure 2 pathogens-10-00828-f002:**
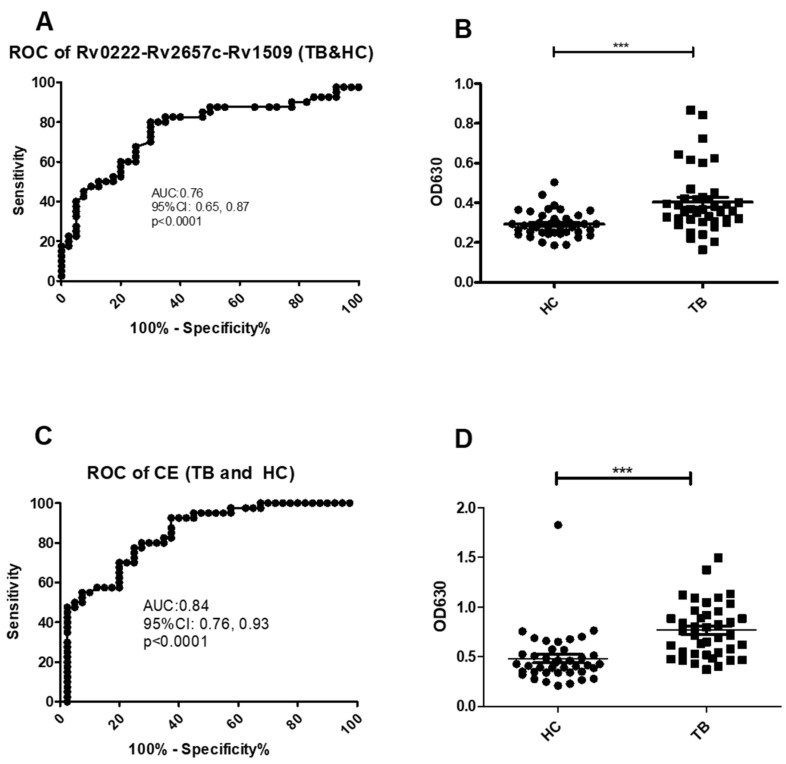
ROC and scatter plots for antibody detection using sera from TB and HC against Rv0222-Rv2657c-Rv1509 and CE. (**A**) ROC of Rv0222-Rv2657c-Rv1509 for TB and HC, (**B**) Rv0222-Rv2657c-Rv1509 antibody detection of TB and HC, (**C**) ROC of CE for TB and HC, (**D**) CE antibody detection of TB and HC. Symbol “***” presents *p* < 0.001.

**Figure 3 pathogens-10-00828-f003:**
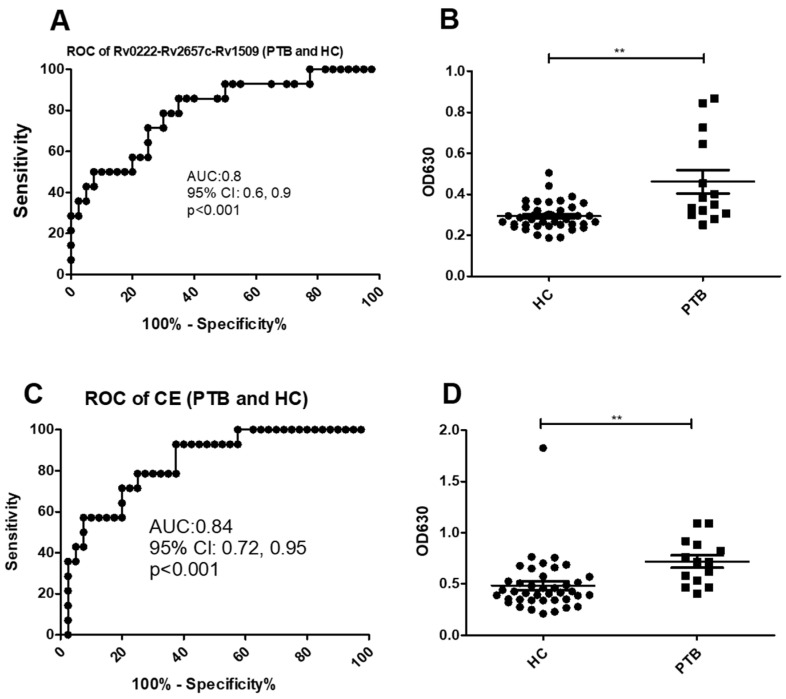
ROC and scatter plots for antibody detection using sera from PTB and HC against Rv0222-Rv2657c-Rv1509 and CE. (**A**) ROC of Rv0222-Rv2657c-Rv1509 for PTB and HC, (**B**) Rv0222-Rv2657c-Rv1509 antibody detection of PTB and HC, (**C**) ROC of CE for PTB and HC, (**D**) CE antibody detection of PTB and HC. Symbol “**” presents *p* < 0.01.

**Figure 4 pathogens-10-00828-f004:**
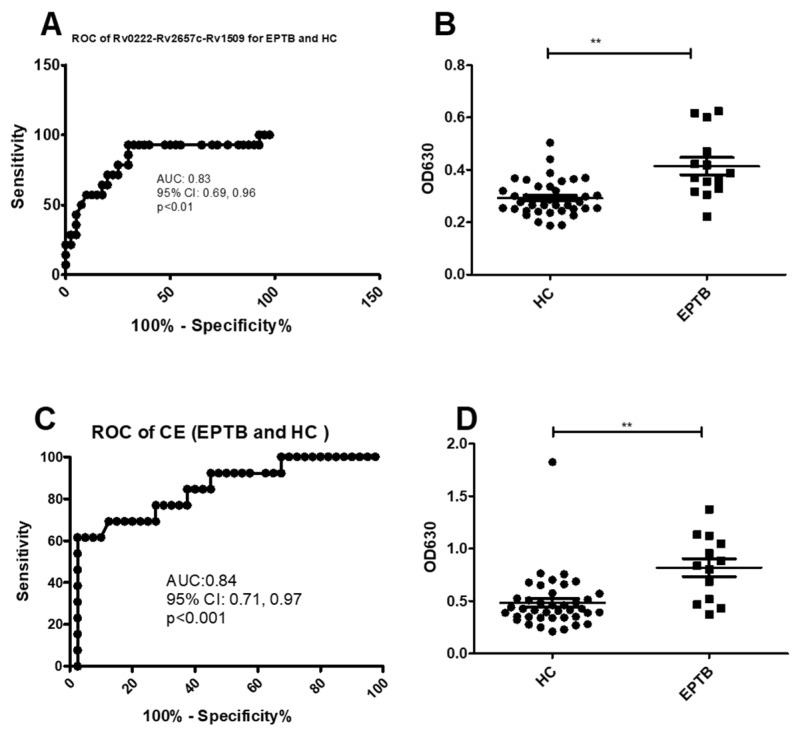
ROC and scatter plots for antibody detection using sera from EPTB and HC against Rv0222-Rv2657c-Rv1509 and CE. (**A**) ROC of Rv0222-Rv2657c-Rv1509 for EPTB and HC, (**B**) Rv0222-Rv2657c-Rv1509 antibody detection of EPTB and HC, (**C**) ROC of CE for ETB and HC, (**D**) CE antibody detection of PTB and HC. Symbol “**” presents *p* < 0.01.

**Table 1 pathogens-10-00828-t001:** A comparison of the diagnostic accuracy of an i-ELISA using either Rv0222-Rv2657c-Rv1509 fusion antigen or a CE.

Antigen	Compared Group *	Sensitivity (%) (95%)	Specificity (%) (95%)
Rv0222-Rv2657c-Rv1509	TB & HC	80 (32/40)	70 (28/40)
		(64.4, 90.9)	(53.5, 83.4)
	PTB & HC	85.7 (12/14)	70 (28/40)
		(57.2, 98.2)	(53.5, 83.4)
	EPTB & HC	92.8 (13/14)	70 (28/40)
		(66.1, 99.8)	(53.5, 83.4)
CE	TB & HC	77.5 (31/40)	72.5 (29/40)
		(61.5, 89.2)	(56.1, 85.4)
	PTB & HC	71.4 (10/14)	80 (32/40)
		(41.9, 91.6)	(64.4, 90.9)
	EPTB & HC	71.4 (10/14)	67.5 (27/40)
		(41.9, 91.6)	(50.9, 81.4)

* TB: tuberculosis; HC: healthy control; PTB: pulmonary tuberculosis; EPTB: extra-pulmonary tuberculosis.

**Table 2 pathogens-10-00828-t002:** Patients’ information.

NO.	Gender	Age	TB Form
1	male	66	PTB
2	male	23	PTB
3	male	60	PTB
4	female	44	PTB
5	male	23	PTB
6	female	45	PTB
7	male	64	PTB
8	male	72	PTB
9	male	77	PTB
10	male	51	PTB
11	male	23	PTB
12	male	68	PTB
13	female	25	PTB
14	male	58	PTB
15	Unknown	31	Unknown
16	Unknown	59	Unknown
17	Unknown	31	Unknown
18	Unknown	37	Unknown
19	Unknown	Unknown	Unknown
20	Unknown	Unknown	Unknown
21	Unknown	Unknown	Unknown
22	Unknown	Unknown	Unknown
23	Unknown	Unknown	Unknown
24	male	18	EPTB
25	female	20	EPTB
26	female	78	EPTB
27	male	41	EPTB
28	female	35	EPTB
29	male	69	EPTB
30	female	Unknown	EPTB
31	female	69	EPTB
32	female	26	EPTB
33	male	53	EPTB
34	female	25	EPTB
35	male	43	EPTB
36	male	34	EPTB
37	female	23	EPTB
38	Unknown	Unknown	Unknown
39	Unknown	Unknown	Unknown
40	Unknown	Unknown	Unknown

**Table 3 pathogens-10-00828-t003:** Primers used in this research.

Gene	Restriction Endonuclease	Primers
Rv0222	*BamH*I	5′-ATA GGATCC ATGAGCAGCGAAAGCGACG-3′ 3′-AGATCCGCCTCCACCTGAACCGCCACCTCCCGCGTAGCCTTCCACCGCAGCA-5′
Rv2657c		5′-GGAGGTGGCGGTTCAGGTGGAGGCGGATCTAGTCTCGGGTGGACGGTC-3′ 3′AGATCCGCCTCCACCTGAACCGCCACCTCCCTCACTGATCGTGATGTACC-5′
Rv1509	*Hind*Ⅲ	5′GGAGGTGGCGGTTCAGGTGGAGGCGGATCTACGGGTGAAGGTTTTGGCAA-3′ 3′GC *AAGCTT* CGAACGCCAGACTCCCTT-5′

The italics indicate restriction endonuclease sites; The underlined part indicates linker.

## Data Availability

Data available in a publicly accessible repository.

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
