# Peer review of "Use of Rv0222-Rv2657c-Rv1509 Fusion Protein to Improve the Accuracy of an Antibody ELISA for Extra-Pulmonary Tuberculosis in Humans"

_pathogens, 2021, doi:10.3390/pathogens10070828_

Round 1

Reviewer 1 Report

This manuscript is certain to be a great interest to a select portion of the biomedical research community. While the paper is limited in breadth, it adds critical diagnostic details regarding one of the world's classic human pathogens.  

The background, methods, findings and data presentation are adequate. The gel figure would benefit from a higher resolution but it is legible and, therefore, adequate, in its current form. 

Author Response

Thank you for your feedback.

Reviewer 2 Report

The research conducted by Chen et al. used Rv0222-Rv2657c-Rv1509 fusion protein as the antigen to detect TB-specific antibody by iELISA method. The study aimed to prove this test showed good performance in diagnosing TB, especially for extrapulmonary TB after testing the serum samples from 40 TB-confirmed patients and 40 healthy controls. The result showed both the sensitivity and specificity of this new antigen were superior to those using CFP10/ESAT6 antigen. The comments for the paper are listed below:

  1. The detailed information of the TB cases and health controls were not addressed. The demographic data, infections sites, interval between onset of symptoms and diagnosis of disease should be described in the paper.
  2. The authors should compared the performance of this test in diagnosing extra-pulmonary TB and pulmonary TB and prove it improves diagnostic accuracy for EPTB in a statistical way. In addition, the results of HC, PTB, EPTB should be put together. Repeated demonstration of the healthy control data in separate figures is redundant.
  3. I am concerned if there is any possibility that cross-reactive antibodies which is not specific for TB is present. The control group should enroll patients with active infection of pathogens other than M. tuberculosis.
  4. There were 12 people without detailed diagnostic information in the TB case group. Were they included in the PTB or EPTB groups for analysis?

Author Response

Point 1: The detailed information of the TB cases and health controls were not addressed. The demographic data, infections sites, interval between onset of symptoms and diagnosis of disease should be described in the paper.

Response 1: Thanks for your suggestion. We listed the information in Table 2. As we focus on TB, PTB and EPTB, we just presented the TB form and gender.

Point 2: I am concerned if there is any possibility that cross-reactive antibodies which is not specific for TB is present. The control group should enroll patients with active infection of pathogens other than M. tuberculosis.

Response 2: Thanks for your suggestion, we added this part of work, and used 5 pneumonia sera for the specific test, they all showed negative in both Rv0222-Rv2657c-Rv1509-iELISA and CE-iELISA. We added in “2.2” Line 86-88.

Point 3: There were 12 people without detailed diagnostic information in the TB case group. Were they included in the PTB or EPTB groups for analysis?

Response 3: Thanks for your suggestion. For the 12 cases, we only knew they were TB patients, but we don’t know if they were PTB or EPTB, so they were not included in PTB or EPTB groups.

Reviewer 3 Report

General comments

This is a well-designed study and well written manuscript that will be of interest to those working with the diagnosis of Tb. The statistical methods appear adequate and the conclusions justified.

The manuscript could be improved by addressing the specific comments below.

Specific comments

L19: replace [Culturing bacilli and Acid-fast Bacillus have a low sensitivity…] with [Isolation of M. tb from culture has a low sensitivity…]

L22: Are you really improving the accuracy or the sensitivity?

L23: you need to include a brief description of the types and numbers of samples involved in the evaluation.

L24: you have defined TB and EPTB and now introduce a third category (PTB) without explanation.

L24-25: You describe sensitivity for TB, PTB and EPTB patients individually, but only one description of specificity. Why?

L26: There has been no previous mention of the CFP10/ESAT6 fusion protein?

L168: A total of 80 samples is very low for this type of evaluation, but you have acknowledged that.

L194: You need to explain what CFP10/ESAT6 are in your introduction.

L195: Why have you selected Rv0222, Rv2657c and Rv1509? You need to explain and justify.

L133: Be careful with the term [accuracy] – are you really referring to accuracy as defined statistically?

L138-9: This description of the antigens should be in the introduction so readers have context for why you are using them.

Author Response

Point 1: L19: replace [Culturing bacilli and Acid-fast Bacillus have a low sensitivity…] with [Isolation of M. tb from culture has a low sensitivity…]

Response 1: Thanks for your suggestion. We replaced the sentence as your suggestion.

Point 2: L22: Are you really improving the accuracy or the sensitivity?

Response 2: Thanks for your question. As we improved both sensitivity and specificity, so totally we improved the accuracy.

Point 3: L23: you need to include a brief description of the types and numbers of samples involved in the evaluation.

Response 3: Thanks for your suggestion. We added “Sera of 40 patients including 14 had PTB, 14 had EPTB and 12 with no information about the form of TB were investigated.”

Point 4: L24: you have defined TB and EPTB and now introduce a third category (PTB) without explanation.

Response 4: Thanks for your suggestion. We explained PTB as “pulmonary TB” in Line 24.

Point 5: L24-25: You describe sensitivity for TB, PTB and EPTB patients individually, but only one description of specificity. Why?

Response 5: Thanks for your suggestion. Specificity is calculated as “ tested negative person that is really health divided by healthy control numbers”, as we had only one ELISA to test the same healthy control, so have only one specificity.

Point 6: L26: There has been no previous mention of the CFP10/ESAT6 fusion protein?

Response 6: Thanks for your suggestion, we added “used as reference antigen” to make it clear.

Point 7: L168: A total of 80 samples is very low for this type of evaluation, but you have acknowledged that.

Response 7: Thanks for your feedback. That’s really limited, so we discussed about that.

Point 8: L194: You need to explain what CFP10/ESAT6 are in your introduction.

Response 8: Thanks for your suggestion. We added “which was regarded as effective antigen for TB diagnosis in many researches” to make it much clearly.

Point 9: L195: Why have you selected Rv0222, Rv2657c and Rv1509? You need to explain and justify.

Response 9: Thanks for your suggestion. We added “Here we found Rv0222, Rv1509 and Rv2657 C were with highly immunogenic and had potential use for TB diagnosis. ” in Line 60 to make it much clearly.

Point 10: L133: Be careful with the term [accuracy] – are you really referring to accuracy as defined statistically?

Response 10: Thanks for your suggestion. Yes here we evaluated accuracy including both sensitivity and specificy.

Point 11: L138-9: This description of the antigens should be in the introduction so readers have context for why you are using them.

Response 11: Thanks for your suggestion. We added “Here we found Rv0222, Rv1509 and Rv2657 C were with highly immunogenic and had potential use for TB diagnosis.” in the introduction to make a brief explanation of those antigens.

Round 2

Reviewer 2 Report

  1. The revised MS included 5 patients with pneumonia other than TB. Please emphasize they are non-TB pneumonia after survey. It would be great If the etiologies of these infections are addressed in the MS.
  2. The title of this article emphasizes the accuracy of iELISA test for extra-pulmonary TB using this novel antigen is improved. In fact, the test showed good performance in diagnosing pulmonary TB as well. The authors should compare the performance of this test with CFP10/ESAT6 in a statistical way. 
  3. N/A always denotes non-applicable. In Table 2, "Unknown" status for some enrolled patients is much suitable.   

Author Response

Point 1: The revised MS included 5 patients with pneumonia other than TB. Please emphasize they are non-TB pneumonia after survey. It would be great If the etiologies of these infections are addressed in the MS.

Response 1: Thanks for your suggestion. I emphasized as “non-TB pneumonia patients sera were used for analytic specific test” in Line 200-201. But sorry we don’t have the etiological results of these patients.

Point 2: The title of this article emphasizes the accuracy of iELISA test for extra-pulmonary TB using this novel antigen is improved. In fact, the test showed good performance in diagnosing pulmonary TB as well. The authors should compare the performance of this test with CFP10/ESAT6 in a statistical way.

Response 2: Thanks for your suggestion. In Line 96-104, we compared the Rv0222-Rv2657c-Rv1509-iELISA with CFP10/ESAT6 in PTB diagnosis in a statistical way.

Point 3: N/A always denotes non-applicable. In Table 2, "Unknown" status for some enrolled patients is much suitable.

Response 3: Thanks for your suggestion. We already changed the “N/A” to “Unknown” in table 2.
